# Anthocyanins in Plant Food: Current Status, Genetic Modification, and Future Perspectives

**DOI:** 10.3390/molecules28020866

**Published:** 2023-01-15

**Authors:** Peiyu Zhang, Hongliang Zhu

**Affiliations:** College of Food Science and Nutritional Engineering, China Agricultural University, Beijing 100083, China

**Keywords:** anthocyanins, genome editing, transgene stacking, transgene free

## Abstract

Anthocyanins are naturally occurring polyphenolic pigments that give food varied colors. Because of their high antioxidant activities, the consumption of anthocyanins has been associated with the benefit of preventing various chronic diseases. However, due to natural evolution or human selection, anthocyanins are found only in certain species. Additionally, the insufficient levels of anthocyanins in the most common foods also limit the optimal benefits. To solve this problem, considerable work has been done on germplasm improvement of common species using novel gene editing or transgenic techniques. This review summarized the recent advances in the molecular mechanism of anthocyanin biosynthesis and focused on the progress in using the CRISPR/Cas gene editing or multigene overexpression methods to improve plant food anthocyanins content. In response to the concerns of genome modified food, the future trends in developing anthocyanin-enriched plant food by using novel transgene or marker-free genome modified technologies are discussed. We hope to provide new insights and ideas for better using natural products like anthocyanins to promote human health.

## 1. Introduction

Anthocyanins are natural water-soluble pigments and are widely distributed in plant food. High anthocyanin accumulation is not only good for extending shelf-life, but also improves the nutrient values of food [1]. Anthocyanins can give radiant color to plants to attract insects, which is beneficial for seed spreading. Additionally, as important secondary metabolites, anthocyanins can also protect plants from biotic and abiotic stress. Thanks to their natural health-promoting activity such as antioxidant and anti-inflammatory, anthocyanins are reported to prevent various chronic diseases such as cardiovascular diseases, neurodegenerative diseases and diabetes [2,3].

In many plant species, anthocyanins act as the major pigments, giving them an attractive color. The anthocyanin biosynthesis pathway is relatively conserved in plants and is regulated by various transcription factors (TFs). A transcription complex formed by MYB, bHLH and WD40 acts as a switch to regulate anthocyanin synthesis. Among them, numerous MYB family members are reported to transcriptional activate anthocyanin biosynthetic genes or repress anthocyanin accumulation in plants food such as apple [4], tomato [5], pear [6], pepper [7], eggplant [8] and sweet potato [9]. In some cases, MYB factors play a vital role to determine the color of food. For example, *VvMYBA1* and *VvMYBA2* are involved in grape skin pigmentation [10]. The origin of the white-skinned grape is associated with the mutation of these two *MYB* loci. *VvMYBA1* is transcriptionally inactivated by the insertion of *Gret1* (a gypsy-type retrotransposon) in its promotor region [11]. Additionally, *VvMYBA2* loses its function due to the single-nucleotide polymorphism (SNP) K980 in the coding sequence [12].

During the past decades, from the perspective of the economic interests of the growers, scientists mainly focused on using genetic engineering to improve the most common resistance traits such as herbicide tolerance [13]. However, with the continuous improvement of quality of life, people are more likely to choose foods that have an attractive appearance such as bright colors and higher nutritional value [14]. Because foods rich in anthocyanins are more popular among consumers, using molecular biotechnology to regulate anthocyanins in plants to improve food quality has become an important topic in related areas.

The traditional breeding process is labor-intensive and has a long production cycle. With the advent of novel transgene and gene editing technologies, people can quickly achieve targeted breeding and create species with targeted traits. An increasing number of genes including regulators and structure genes that involve in the anthocyanin biosynthesis pathway are being annotated; they can be the candidates in anthocyanin molecular regulation [15,16].

Plant genome modification is mainly achieved by *Agrobacterium*-mediate or biolistic transformation, and the transgene components are integrated into the host genome. To ensure the generations are transgenic, the inducing and regeneration process is usually accompanied by an antibiotic selection [17]. Concerns about the consumption of transgenic plant food with transgene constructs have been raised [18]. To address these concerns, some international organizations or governments of various countries have adopted legislation to implement biosecurity management. Although transgenic crops undergo a pre-market regulatory safety assessment in many countries, the regulatory requirements for transgenic products may be different [19]. Additionally, the current regulations, especially for the management of gene editing crops, may be imperfect in some nations [20]. Considering that modern biotechnology breeding is usually limited by the various regulatory systems, transgene/marker-free biotechnologies are promising tools to produce genome modified plant food with desired modification. In response to this trend in development, various transgene- or marker-eliminating methods have been established [21]. Here, we summarized the current status of anthocyanins in plant food, the recent advances in the molecular regulation of anthocyanin, and discussed the future trends in developing anthocyanin-enriched plant food by using novel transgene- or marker-free genome modified technologies.

## 2. Anthocyanins in Plant Food

### 2.1. Chemical Structure of Anthocyanins

Anthocyanins share the basic diphenyl propane skeleton of flavonoids (C6C3C6), formed by two aromatic rings connected to a benzene ring (Figure 1A). The natural anthocyanins are subjected to hydroxylation, methylation, glycosylation and acylation modifications, which not only resulted in color variation but are also beneficial for the stability of these compounds [22]. As the number of hydroxyl groups increased, the anthocyanin blue color deepened, while methylation contributed to a red shift and stabilization of anthocyanins [23,24]. The glycosylation of anthocyanin increases its stability for storage in the vacuole. Glucose, galactose, rutinose, rhamnose, arabinose and xylose are the most common sugars that exist in anthocyanins. The glucosides tend to attach to anthocyanidins as mono- or polysaccharides at the C5- or C7-position of the A-ring or the C3-position of the C-ring (Figure 1A) [22]. These sugars can be acylated further with various aliphatic or aromatic acids, contributing to the diversity of natural anthocyanin compounds. Although over 700 natural anthocyanins and 27 anthocyanidins have been identified, only a few kinds of anthocyanidins including cyanidin, delphinidin, pelargonidin, peonidin, malvidin and petunidin are common in foods [25]. The color of anthocyanins is associated with the pH value: they are red at low pH value (pH 1–3), but become yellow or colorless at pH 4–5 and turn to blue or purple at pH 6–7 [26].

### 2.2. Anthocyanins Exist in Common Plant Food

The variety and content of anthocyanins varied among different plant foods (Table 1) but are predominantly found in berry fruits. Blueberry is welcomed by consumers for its high total anthocyanin content, reaching 1202 mg cyanidin-*O*-glucoside equivalents kg^−1^ fresh fruit [27]. Additionally, blackberry (*Rubus*) and blackcurrant (*Ribes nigrum*) fruits are also characterized by high concentrations of anthocyanins reaching 94.76 and 113.79 mg 100 g^−1^ fresh weight. Cyanidin-3-*O*-glucoside accounts for 93% of all anthocyanins detected in blackberry, while delphinidin-*O*-glucoside and delphinidin-*O*-rutinoside represent dominant anthocyanins in blackcurrant [28]. Anthocyanins mainly exist in grape fruit skin. In some red-skinned grape cultivars (*Vitis vinifera*) the average total content of anthocyanin is 120 mg 100 g^−1^ fresh weight, and four compounds including delphinidin-3-*O*-glucoside, malvidin-3-*O*-glucoside, petunidin-3-*O*-glucoside and malvidin-3-*O*-glucoside-5-*O*-glucoside account for almost 60% of the total anthocyanin content [29]. Black rice (*Oryza sativa*) is an important staple food that produces anthocyanins, and at least 18 kinds of anthocyanins are detected, during which the cyanidin-3-*O*-glucoside and peonidin-3-*O*-glucoside are the dominant compounds. In the ‘Niaw Dam Pleuk Khow’ variety, the average anthocyanin content can reach 442 mg 100 g^−1^. By comparative quantification of fourteen purple corn (*Zea mays*) samples, the average anthocyanin content ranges from 430 to 11,700 mg 100 g^−1^ powders, and six anthocyanins including cyanidin-3-glucoside, pelargonidin-3-glucoside, peonidin-3-glucoside, cyanidin-3-*O*-(6″-malonylglucoside), pelargonidin-3-*O*-(6″-malonylglucoside) and peonidin-3-*O*-(6″-malonylglucoside) are detected as the major compounds [30].

## 3. Biosynthesis and Regulatory of Anthocyanins in Plant Food

### 3.1. Structure Genes Involved in Biosynthesis Pathway

Anthocyanins are one of the end products of the flavonoid synthesis pathway [35]. The biosynthesis of anthocyanins in higher plants is a relatively conserved process and has been extensively reported (Figure 1A). The first stage is a common step of the flavonoids biosynthetic pathway, during which the initial substrate phenylalanine is catalyzed by phenylalanine ammonia lyase (PAL), cinnamate 4-hydroxylase (C4H) and Cinnamic acid-4-hydroxylase (4CL), then converted to coumaroyl CoA [25]. The second stage is the metabolism of flavonoids to produce anthocyanin precursors dihydroflavonols. Chalcone synthase (CHS), chalcone isomerase (CHI), flavanone-3-hydroxylase (F3H), flavanone 3′-hydroxylase (F3′H) and flavanone 3′5′-hydroxylase (F3′5′H) involved in this early synthetic stage of anthocyanins [36]. Subsequently, catalyzed by dihydroflavonol 4-reductase (DFR), various leucoanthocyanidins will be formed. Additionally, then anthocyanidin synthetase (ANS) converts colorless leucoanthocyanidins to anthocyanidins. In plant cells, anthocyanidins tend to bind to glycosides and exist in anthocyanin glycosides form catalyzed by flavonoid 3-*O*-glucosyltransferase (UFGT). The genes involved in the early common flavonoid biosynthetic pathway are called *early biosynthetic genes* (*EBGs*) and the others including F3′5′H, DFR and ANS are mainly responsible for the formation and accumulation of anthocyanins are called *late biosynthetic genes* (*LBGs*) [25]. Once synthesized in the cytosol, anthocyanins will be transported into vacuoles by glutathione S-transferase (GST) [37].

### 3.2. Other Regulators of Anthocyanin Biosynthesis

In plants, apart from the structure genes, related transcription factors, various environmental factors and internal phytohormones also affect anthocyanin biosynthesis (Figure 1B).

#### 3.2.1. Transcriptional Factors Involved in Anthocyanin Biosynthesis

Although there are kinds of regulatory genes that influence anthocyanins synthesis and metabolism, the key *LBGs* involved in the biosynthesis of anthocyanins are mainly regulated by conserved transcriptional factors complex [38]. The regulatory complex comprises a R2R3-MYB transcription factor, a subgroup lllf basic Helix-Loop-Helix (bHLH) transcription factor and a WD-repeat protein (WDR) to form a ternary MBW complex [16]. The bHLH and WD are usually constitutive expression in plants, while various MYB members are reported to play different roles in anthocyanins biosynthesis pathway (Table 2) [15,39].

By investigating different varieties and hybrids with different colors of potato tubers and leaves, three *R2R3 MYBs* (*StAN1*, *StMYBA1* and *StMYB113*) and two bHLHs (*StbHLH1*, *StJAF13*) and *StWD40* are considered as the regulatory genes controlling anthocyanin biosynthesis [40]. Among them, *StAN1* is the predominant candidate gene in regulating the biosynthesis of anthocyanin [41,42]. In sweet potato (*Ipomoea batatas*), a novel complex IbERF71-IbMYB340-IbbHLH2 is characterized and is found to promote anthocyanins accumulation by binding to the promotor of *IbANS1* [43]. More recently, *IbMYB1* is a necessary regulator which constitutes three members, *IbMYB1-1*, *IbMYB1-2a* and *IbMYB1-2b*. *IbMYB1-2* has also been recognized as a master regulatory gene activating the expression of *LBGs* [44]. *IbMYB44* is suggested to be a repressor in anthocyanin biosynthesis. IbMYB44 can interact with IbbHLH2, IbNAC56a or IbNAC56b, competitively inhibiting the IbMYB340-IbbHLH2-IbNAC56 regulatory complex formation [9]. Wild species producing anthocyanin-cultivated common species of tomato fruit is usually anthocyanin-free, while some wild lines, such as ‘Indigo Rose’ (InR), exhibit a purple skin color because of the accumulation of anthocyanins. This natural diversity of tomato species provides an ideal material to investigate the regulatory network of anthocyanin biosynthesis in plants. LnR fruit has both an *Aft* (*Anthocyanin fruit*) and an *atv* (*atroviolacium*) locus. There are four putative candidate R2R3-MYB genes at the *Aft* locus: *SlAN2* (*SlMYB75*), *SlANT1* (*Slanthocyanin 1*), *SlANT1like* (*SlMYB28*) and *SlAN2like* (*SlMYB114*). The *atv* locus encodes a repressor SlMYBATV, which belongs to the R3-MYB factors and has lost its function in lnR fruit [5]. SlANT1 and SlAN2 in tomatoes have been proven to be positive regulators in the production of anthocyanin [45,46]. By conducting a comparative functional analysis of the four Aft locus candidate genes, *SlAN2like* shows the strongest ability in transcriptional activating the expression of anthocyanin biosynthesis genes and related regulatory genes. So, SlAN2like plays a vital role in promoting anthocyanin accumulation in tomatoes [5,47]. However, in domesticated species, alternative splicing results in shorter transcripts of *SlAN2like*, thus producing a premature protein. The truncated protein loses the R3 domain, which contains the bHLH-binding site, and therefore fails to associate with bHLH partners. Finally, the MBW regulator complex cannot form because of the absence of R2R3-MYB factor SlAN2like, and then, the anthocyanin biosynthesis related genes, especially the *LBGs* such as *DFR*, cannot be transcriptionally activated [5,47]. The negative regulator SlMYBATV competes with Aft to associate with SlJAF13 (bHLH1) and SlAN1 (bHLH2), inhibiting the formation of the MBW complex and thus negatively regulating anthocyanin biosynthesis in tomato fruit [5,48].

Except for the core transcription factors involved in forming the MBW complex, a huge number of other transcriptional factors such as WRKY are also involved in the regulation of anthocyanin biosynthesis or metabolic pathway, especially under external stimulation or stress such as light, drought, or phytohormone (Table 2).

#### 3.2.2. Phytohormones

Ethylene as one of the most important phytohormone during plant development also affects anthocyanins contents. However, a large amount studies have revealed that the effects of ethylene on anthocyanin biosynthesis vary among plant species. Ethylene has been demonstrated in apple to have positive effects on anthocyanin synthesis. MdMYB1 is responsible for the biosynthesis of anthocyanins in apple fruit, and MdEIL1 binds to the promotor of MdMYB1 to promoting the accumulation of anthocyanins. MdMYB1 binds to an ethylene metabolism pathway related gene MdERF3 conversely, resulting an increased ethylene production [49]. Ethylene inhibits anthocyanin biosynthesis mainly by repressing anthocyanin-related genes including a key R2R3-MYB factor *SlAN2-like* in tomato [50]. In pear, a transcriptional activator PpERF105 could activate the expression of an R2R3-MYB repressor PpMYB140, which inhibits the expression of structure genes involving in the anthocyanin biosynthesis [51].

Apart from ethylene, other phytohormones also affect anthocyanin biosynthesis in plants. ABSCISIC ACID-INSENSITIVE5 (ABI5) is a crucial regulator of ABA signaling, and MdABI5 interacts with MdbHLH3 to positively regulate MdDFR and MdUF3GT, promoting anthocyanin biosynthesis in apples [52]. Exogenous salicylic acid treatment can considerably promote the biosynthesis of anthocyanins in grapes [53].

#### 3.2.3. Temperature

The temperature is an environmental factor affecting anthocyanin biosynthesis and metabolism, and generally, low temperature is beneficial for the accumulation of anthocyanins in plants. The changes of temperature alter the expression of transcriptional factors involved in anthocyanin biosynthesis. In apples, a hot climate and artificial heating resulting a dramatic decrease of anthocyanins in fruit peel. At the same time, the expression of related transcriptional activation genes, especially an R2R3 MYB (*MdMYB10*), is rapidly reduced [54]. In potatoes, high temperature results in a reduction of anthocyanins in tuber skin and flesh. Two negative R2R3 MYB TFs, *StMYB44-1* and *StMYB44-2*, are up-regulated under high temperature, resulting in decreased expressions of the R2R3 MYB TFs *StAN1* and *StbHLH1* [55].

#### 3.2.4. Light Signal

Light is an essential factor during plant growth and development, and the plant anthocyanins biosynthesis process is usually induced by light [56,57,58]. A bZIP TF HY5 plays a crucial role in anthocyanin biosynthesis in response to light. HY5 can bind to the promotor of various genes involved in light signal transduction and anthocyanin biosynthesis [59,60,61]. For example, SlHY5 binds the ACE-box in the promotor region of *SlAN2like^lnR^* to enhance anthocyanin production [5]. However, HY5 and MYB TFs can be ubiquitinated in dark-grown seedlings by the CONSTITUTIVE PHOTOMORPHOGENIC 1/SUPPRESSOR OF PHYTOCHROME A-105 (COP1/SPA) E3 ubiquitin ligase, and causing the degradation of HY5 and MYB factors [62,63]. However, when the plants are exposed to light, the activated photoreceptors rearrange the COP1/SPA complex to make it non-functional, thus improving the stability of HY5 or other positive regulators in anthocyanin biosynthesis pathway such as MYB factors [64].

**Table 2 molecules-28-00866-t002:** Transcription factors involved in anthocyanin biosynthesis.

Species	Gene	Type	Function	Mechanisms	Reference
*Vitis vinifera*	*VvMYB86*	R2R3-MYB	-*	*VvMYB86* represses anthocyanin biosynthesis branch in grapes by downregulating the transcript levels of *VviANS* and *VviUFGT.*	[65]
	*VvMYB2r*	R2R3-MYB	+	VvMYBA2r along with VvMYCA1 and VvWDR1 form the MBW complex, which could activate the promoter of *VvUFGT* gene, promoting anthocyanin accumulation of grape skin.	[66]
*Citrus sinensis*	*CsRuby1*	R2R3-MYB	+	*CsRuby1* encodes a MYB transcription factor that serves as the key positive regulator of anthocyanin biosynthesis.	[67]
*Ipomoea batatas*	*IbMYB44*	R2R3-MYB	-	IbMYB44 interacts with IbbHLH2, IbNAC56a or IbNAC56b, competitively inhibiting the IbMYB340-IbbHLH2-IbNAC56 regulatory complex formation.	[9]
*Solanum tuberosum*	*StWRKY13*	WRKY TFs	+	*StWRKY13* enhances the role of *StAN2* in promoting anthocyanin biosynthesis in tobacco. StWRKY13 interacts with the promotor of *StCHS*, *StF3H*, *StDFR*, and *StANS* to enhance their activities.	[68]
*Solanum lycopersicum*	*SlJAF13*	bHLH	+	SlJAF13 takes part in the first MBW complex and activates *SlAN1*. Additionally, SlJAF13 interacts with SlMYC2, inhibiting SlMYC2 activation of *SlJAZ2* transcription.	[69]
	*SlAN1*	bHLH	+	SlAN1 takes part in the second MBW complex and transcriptional activates the expression of anthocyanin biosynthesis related genes.	[70]
	*SlAN2 (SlMYB75)*	R2R3-MYB	+	*SlAN2* acts as a positive regulator under high light or low temperature. SlAN2 can directly bind to the *MYBPLANT* and *MYBPZM* cis-regulatory elements and to transcriptional activate the *LOXC*, *AADC2* and *TPS* genes.	[71]
	*SlANT1*	R2R3-MYB	+	*SlANT1* up-regulates structural genes in the anthocyanin pathway.	[70]
	*SlAN2like*	R2R3-MYB	+	*SlAN2like* activates the expression of anthocyanin biosynthetic genes and related regulatory genes.	[5]
	*SlMYBATV*	R3 MYB	-	SlMYBATV competes with Aft for binding to bHLHs, and negatively regulates anthocyanin biosynthesis.	[5]
*Pyrus pyrifolia*	*PybHLH64*	bHLH	+	PpbHLH64 interacts with PpMYB10 to form an MBW complex.	[72]
	*PyMYB10, PyMYB10.1*	R2R3-MYB	+	PpMYB10 and PpMYB10.1 interacts with PpbHLH to form the MBW complex.	[6]
*Pyrus*	*PpMYB140*	R2R3-MYB	-	PpMYB140 acts as a competitor that competes with PpMYB114 to form the MBW complex.	[51]
	*MdWRKY41*	WRKY TFs	-	MdWRKY41 downregulates the expression of *MdMYB12*, and interacts with MdMYB16 to form a complex, which suppresses the expression of *MdANR* and *MdUFGT*.	[73]
	*MdbZIP44*	bZIP TFs	+	MdbZIP44 binds to MdMYB1 in response to ABA and upregulates downstream target genes to promoting anthocyanin accumulation.	[74]
	*MdNAC52*	NAC	+	MdNAC52 binds to the promoters of *MdMYB9* and *MdMYB11* to promote anthocyanin accumulation.	[75]
	*MdERF38*	ERFs	+	*MdERF38* can in response to drought stress and interacts with MdMYB1 to positively regulate anthocyanin biosynthesis.	[76]
	*MdMYB3*	R2R3 MYB	+	*MdMYB3* involves in transcriptional activation of several flavonoid pathway-related genes to enhance the skin color of fruits.	[4]
	*MdMYC2*	bHLH	+	MdMYC2 interacts with a Jasmonate signaling pathway repressor MdJAZ2, upregulating the expression of downstream genes such as *MdDFR*, *MdUF3GT*, *MdF3H* and *MdCHS*.	[77]
	*MdMYB15L*	MYB TFs	-	MdMYB15L interacts with MdbHLH33 and weakens MdbHLH33-induced anthocyanin accumulation.	[78]
*Actinidia chinensis*	*AcMYBF110*	R2R3-MYB	+	AcMYBF110 participates in the formation of the AcMYBF110-AcbHLH1-AcWDR1 complex to induce anthocyanin accumulation.	[79]
*Fragaria × ananassa*	*FaRAV1*	RAV TFs	+	*FaRAV1* up-regulates *FaMYB10* and the genes involved in phenylpropanoid and flavonoid biosynthesis pathway.	[80]
	*FaBBX22*	B-box TFs	+	*FaBBX22* positively regulates anthocyanin biosynthesis by enhancing related genes (*FaPAL*, *FaANS*, *FaF3′H*, *FaUFGT1*) and transporting gene *FaRAP* in a light-dependent manner.	[81]
*Brassica rapa*	*BrMYBL2.1*	R3 MYB	-	*BrMYBL2.1-G* (from a Chinese cabbage cultivar with purple leaves) represses transcriptional activation of *BrCHS* and *BrDFR* via blocking the activity of the MBW complex.	[82]
*Daucus carota*	*DcMYB6*	R2R3 MYB	+	DcMYB6 contains the conserved bHLH-interaction motif and two typical motifs of anthocyanin regulators.	[83]
*Brassica oleracea*	*BoMYB2*	R2R3 MYB	+	BoMYB2 interacts with various BobHLHs to form the MBW complex, and positively regulates the *LBGs* in anthocyanin biosynthesis.	[84]
*Solanum melongena*	*SmMYB35*	R2R3 MYB	+	SmMYB35 interacts with SmTT8 and SmTTG1 to form a MBW complex, and positively regulates *SmCHS*, *SmF3H*, *SmDFR*, and *SmANS*.	[8]
*Oryza sativa*	*OsTTG1*	WD40	+	OsTTG1 encodes a WD40 protein, and interacts with Kala4, OsC1, OsDFR and Rc.	[85]
	*OsKala4*	bHLH	+	*Kala4* involved in the origin of black rice corresponds to *Os04g0557500*, which encodes a bHLH transcriptional factor. A structural change in the *OsKala4* promoter induced ectopic expression of this bHLH protein, thus resulting in the birth of black rice.	[86]
*Zea mays*	*Zmp1, Zmp2*	R2R3 MYB		*Zmp1* (*ZmMYB3*) *and Zmp2* (*ZmMYB55*) encode R2R3-MYB transcription factors that accumulates flavonoid such as 3-deoxyflavonoids, flavones, and phlobaphenes.	[87,88]
	*Zmc1, Zmpl1*	R2R3 MYB		*Zmc1* (*ZmMYB1*) and *Zmpl1 (ZmMYB2)* function in the MBW complex and upregulate *LBGs* expression.	[87,88]

*: ‘+’ and ‘-’ represents positive and negative regulation on anthocyanin accumulation, respectively.

## 4. Genetic Engineering to Produce Anthocyanin-Enriched Plant Foods

Although anthocyanins are widely distributed in plant food, extraction of products from these food matrices directly does not guarantee high yield and efficient production. Thus, fast and eco-friendly innovative methodologies and technologies can be used to regulate phytonutrients such as anthocyanins. Generally, there are two basic strategies to enrich selected nutrients in plants using biotechnologies, that is, promoting its biosynthesis or inhibiting the metabolism branch road. Anthocyanin biosynthesis and metabolism pathway are well understood, and the genes involved in the related branch road have been identified, which provides the basis for scientists to regulate anthocyanin in plant food. Altering the biosynthesis and metabolism related gene expression is the most direct way, where the transgene stacking and gene editing technologies can be used.

### 4.1. Genetic Engineering Techniques

Introducing positive foreign genes or silencing endogenous negative factors is the major method of genetic engineering for crop improvement. *Agrobacterium tumefaciens*-mediated transformation is the most widely used tool to introduce foreign genes into plant cells, in which the Ti plasmid DNA carrying the target gene can insert into the host genome stably [89].

Those genes that come from other species are introduced into plant cells, giving the host new traits such as anti-herbicide. The first transgenic crops are tobacco and petunia, which were produced in 1983 [90,91]. In the past decades, more and more transgenic plants emerged, such as maize, cotton, potato, soybean, and others [92]. The data suggested that by using transgenic technology, crop yields have increased by 22%, leading to a considerable increase of growers [93]. Recently, a strategy for introducing more than one gene into plant cells, which is called transgene stacking, has been developed [94]. Usually, only a single gene is transformed into selected plant cells to achieve genome modification in a traditional transgene system. However, multigene integration is more effective and becomes a promising tool in breed improvement, such as anthocyanin regulation [95,96].

Unlike transferred foreign genes, classical genome editing technology is mainly used to produce loss-of-function crops. The clustered, regularly-interspaced, short-palindromic repeats (CRISPR) system is ubiquitous in bacteria for resistance against biotic stresses and was discovered in 1987 [97]. Over the last 10 years, the CRISPR system has been rapidly developed and widely applied for plant genome editing. Cas endonuclease and sgRNA (small guide RNA) are the core members of the system. Cas9 nuclease recognizing NGG PAM (protospacer adjacent motif) is the most common in plant genome editing, while other types of Cas such as Cas12a (Cpf1), Cas12b (C2c1) and Cas12e (CaxX) have flexible PAM recognition sites and are promising tools in producing longer deletions [98]. In addition to producing loss-of-function gene editing events, the novel CRISPR/Cas systems based on catalytically dead Cas (dCas) has been established and applied in gain-of-function research. Because of the mutations in the nuclease domain, the dCas protein cannot cut DNA to induce double-strand breaks (DSB) while the RNA-guided DNA binding competent ability is retained. Being fused to effectors and assembled with multiple sgRNAs, the multigene can be activated and enhanced [99,100,101].

Although there have been a few transgenic commercial products before [102,103,104], the emergence of the CRISPR market has been developed recently. A successful example is the γ-aminobutyric acid (GABA)-rich tomatoes created in 2017, which has entered the market in Japan [105]. Scientists using CRISPR/Cas9 technology to delete a C-terminal autoinhibitory domain of SlGAD2 and SlGAD3, thus improving glutamate decarboxylase activity and increasing GABA content in tomato fruit [106]. Recently, the USDA approved a transgenic purple tomato, which was produced by overexpression of two anthocyanin synthesis related genes from the snapdragon. The agency announced that “this plant may be safely grown and used in breeding in the United States” and is not subject to regulation (https://www.aphis.usda.gov/aphis/newsroom/stakeholder-info/sa_by_date/sa-2022/purple-tomato (accessed on 3 January 2022)).

### 4.2. Anthocyanins Improved Transgenic Crops

#### 4.2.1. Tomato

Tomato is an important economic crop with an abundance of nutrients. However, artificial domestication leads to reduced species diversity. Compared with wild species, cultivated tomatoes lose some potential good traits, for example, the common cultivars do not synthesize anthocyanins. In consideration of the high nutritional value of anthocyanins, various attempts have been down to produce anthocyanin-enriched tomatoes. At an early stage, overexpression of selected *EBGs* such as *CHI* failed to enrich anthocyanins in tomatoes, although the total flavonoid content was improved [107,108,109]. Until an MYB-factor *SlANT1* was overexpressed under a strong cassava vein mosaic promoter, the transgenic tomato fruit peel showed purple spots. At the same time, the overexpression of *SlANT1* up-regulated the *EBGs*, *LBGs*, and the genes involved in the anthocyanin glycosylation and transport into the vacuole [110]. The first successful artificial anthocyanin-enriched tomato fruit was produced in 2008. Here, two regulatory genes *AmDelila* and *AmRosea1* from the snapdragon (*Antirrhinum majus*), were overexpressed in tomato fruit under the fruit-specific promotor *E8*. *AmDel* encodes a bHLH factor and *AmRos1* encodes an MYB factor. The highest concentration of anthocyanins was detected in a selected line N, reaching 283 mg 100 g^−1^ FW [111]. As compared to the cultivar LnR, which introgressed from *Solanum chilense* and *Solanum cheesemanii*, transgenic tomatoes showed a completely purple color both in the peel and pulp, while LnR only accumulated anthocyanins in the part of fruit skin that was exposed to sunlight [112]. Recently, overexpression of a single gene *SlAN2* (*SlMYB75*) by using a 35S promoter also produced a transgenic tomato that had anthocyanins reaching 186 mg 100 g^−1^ FW. And some other genes such as *SlLOXC*, *SlAADC2,* and *SlTPS* involved in downstream metabolic pathways were enhanced with the overexpression of SlMYB75 [71].

#### 4.2.2. Rice

Rice is one of the staple crops in the world. Using genetic engineering techniques to improve the nutrients in rice is of great importance. For example, by introducing the β-carotene biosynthetic pathway into rice endosperm, scientists obtain the so-called “Golden Rice”, which has 0.16 mg 100 g^−1^ carotenoid in the endosperm. This germplasm improvement is beneficial to deal with vitamin A deficiency in rice-feeding countries [113]. To achieve the biosynthesis of anthocyanin in rice endosperm, a high-efficiency multi-transgene stacking vector system, TransGene Stacking II (TGSII) was established. The system was based on the Cre recombinase/loxP-mediated recombination to assemble multiple genes. Here, eight anthocyanin biosynthesis related genes including two regulatory genes *ZmLc* (bHLH TF) and *ZmPl* (R2R3-MYB TF) from maize, and six structural genes *SsCHS*, *SsCHI*, *SsF3H*, *SsF3′H*, *SsDFR* and *SsANS* from *Coleus* were assembled. These genes were introduced into rice under a series of rice endosperm-specific promotors from rice. In the “Purple Endosperm Rice”, the highest content of anthocyanins could reach ~100 mg 100 g^−1^ of dried grain, and the antioxidant activities are improved as compared to the colorless lines [95].

#### 4.2.3. Maize

In addition to its health benefits, maize rich in anthocyanins can also be used to replace blueberry berries as raw materials for health products, as well as in food, beverage and cosmetics industries, which can greatly reduce economic costs. Therefore, it has a broad market prospect to use biotechnology to create maize rich in anthocyanins. In 2018, the first embryo and endosperm anthocyanin-rich purple maize was produced by overexpression of four genes including *ZmBz1*, *ZmBz2*, *ZmC1* and *ZmR2* simultaneously under the control of modified tissue specific promoters. The total anthocyanin content of the transgenic maize reached 291 mg 100 g^−1^ [114].

#### 4.2.4. Other Species

*StISAC* is proven to be the direct ortholog of *SlMYBATV-like*. *StISAC* negatively regulates anthocyanins biosynthesis by recruiting bHLH co-partners, inhibiting the formation of MBW complex. So, in the potato variety ‘Blue Star’, CRISPR-Cas9 system is used to knock out *StISAC*. In the mutant cell lines, the content of anthocyanins was doubled, and the pigmentation was stabilized over time [115].

In grapevine, a bZIP gene *VvbZIP36* belonging to group K is proven to be a negative regulator of anthocyanin biosynthesis. Knocking out one allele of *VvbZIP36* in grapevine results in the accumulation of flavonoids and anthocyanins compounds such as chalcone, naringenin, dihydroflavonols and cyanidin-3-*O*-glucoside [35].

Recently, a third-generation CRISPRa system, CRISPR-Act3.0, is applied in pear to regulate anthocyanin biosynthesis. Here, AtUBQ10 and AtU3 promotor are chosen to express dCas9 and sgRNAs. An activation domain VP64 is fused with the dCas9, and two or four sgRNAs targeting to the promotors of *PybZIPa*, *PyMYB10*, *PyMYB114*, *PybHLH3*, *PyDFR*, *PyANS* or *PyUFGT* are assembled simultaneously. By using this strategy, several genes are upregulated at least 10-fold. Finally, the CRISPR-Act3.0 edited lines has a strong accumulation of anthocyanins and appeared red in color [116].

## 5. Future Perspective: Transgene/Marker-Free Anthocyanin Improved Crops

Thousands of studies have been done to investigate the molecular mechanism of the anthocyanin synthesis pathway, and numerous structure genes and regulatory factors have been identified. Based on these remarkable findings, genetic engineering technologies such as multi-gene ectopic expression or gene editing methods can be used to improve the existing germplasm resources. Several varieties rich in anthocyanins have been successfully produced. Notably, the “big purple tomato” designed in 2008 has been approved to be safely grown and used in breeding and may be released to American stores in 2023.

However, there are always debates about the genome-modified food, and the legislative issue in the field is still an urgent one to be solved. Insertion of foreign genes resulted in gene flow between different species, and the potential toxicity and allergenicity from transgenic marker genes or resistance genes to humans is the biggest concern for consumers [92]. To solve this problem, transgene-free strategies are developed. There have been several approaches to producing transgene or marker-free genome modified plants:

(1) Eliminating foreign sequences through genetic segregation: the genome modified components is usually introduced into plant cells by Agrobacterium-mediated or biolistic method. The sequences on the Ti plasmid are integrated into plant genomes. In most cases, T0 plants are heterozygous and with transgenes. According to Mendelian genetics, about 25% T1 generation segregating from T0 plants will be transgene-free and the modified locus will be homozygous if there is only one copy of the transgenes. However, use of the common Agrobacterium-mediated or biolistic method tends to result in more than one copy insertion, so the actual probability to get the desired transgene-free modified plants is much lower. Genetic segregation is the most common way to get transgene-free modified plants in current status (Figure 2A) [117,118].

(2) Fluorescence tag-assisted screening: it is unrealizable to isolate transgene-free modified plants in T0 generation directly, so it is time-intensive using a traditional procedure to screen desired progeny. Introducing a fluorescence marker such as mCherry in plant cells makes it possible to identify the transgene-free seeds of T1 plants. Thus, only the transgene-free seeds are selected to grow for further genome-modified analysis. Although this method is not limited by specific species and has broad spectrum applicability, results suggested that only a few mutations generated in T1 generation can be transmitted to progeny (Figure 2B) [117,119,120].

(3) Using transient expressing editor without antibiotic selection: genome modified constructs can be transiently expressed in plant cells by Agrobacterium and will not be integrated into the host genome. The transient expression vector does not include antibiotic selection units, so three populations exist in T0 plants: plants having foreign gene insertion, untransformed plants, and transiently transformed plants. So, it is time-consuming to identify transgene-free progeny with desired traits (Figure 2C) [117,121].

(4) Preassembled CRISPR/Cas9 ribonucleoproteins (RNPs): Cas9 protein is expressed and sgRNA is transcribed in vitro. Moderate Cas9 protein and sgRNA are preassembled in vitro, and the ribonucleoprotein complex is used to transfect protoplast or callus, which regenerate into seedlings later. Because the RNP complex does not include a resistance screening component, the cells and calluses with or without RNP component can regenerate into seedlings. Although all T0 plants are transgene-free, a few plants are edited or modified (Figure 2D) [117,122,123].

(5) Using suicide Transgene Killer CRISPR (TKC) system: two suicide transgenes cassettes are assembled onto the genome modified vector in the TKC system. A toxic protein encoding gene BARNASE is expressed and driven by an early embryo-specific promotor REG2. Another rice male gametophyte specific lethal gene CMS2 is driven by a constitutively strong promoter CaMV 35 s. Once the callus passes through the period of vegetative growth, the transgene-containing pollen will be eliminated by CMS2 in the stage of reproductive growth, and the embryos containing the transgene constructs will be killed by BARNASE. Thus, transgene constructs are eliminated automatically in the genome modified progeny by using the TKC system. However, differences in the function of specific lethal genes in different species such as dicotyledons and monads need to be considered in this approach (Figure 2E) [124].

(6) Using a multi-transgene stacking vector system accompanied by a marker-free donor vector: a marker excision donor vector is constructed in this system. Two cassettes are assembled onto this plasmid between the Gateway *attB1/B2* sites: one is the Cre recombinase driven by a pollen-specific promotor *Pv4*, and another is the resistance gene cassette. Additionally, there is a *loxP* site on the inner side of the *attB1* or *attB2* site. On the transgene vector, a lethal gene *ScaB* in *E. coli* in the presence of sucrose is located between the Gateway *attP1/P2* sites. Before transfection, the transgene plasmid and marker excision plasmid are mixed and the *Cre* and resistance gene cassettes are assembled onto the transgene plasmid by Gateway BP reaction. Because Cre recombinase only exists in pollen, callus can be inducted and regenerate accompanied by an antibiotic screening. At the reproductive growth stage of T0 plants, the resistance gene and *Cre* cassette in male gametophytes will be eliminated by Cre-mediated recombination. Thus, T1 plants are heterozygosis. After selfing and segregation, marker-free T2 progeny can be selected (Figure 2F) [95].

Although various transgene/marker-free methods have been established, application in plant food anthocyanins modification is rare. Although transgene constructions can be eliminated by using these methods, the efficiency of most methods is extremely low to screen the transgene-free plants with desired modification. Additionally, species-specific genes used in several methods are another limiting factor that affects the popularization of these methods. It’s an inevitable trend to produce transgene/marker-free anthocyanin-enriched plant food in the future, but further modifications of these methods are needed to improve screening efficiency or broaden their applicability across species.

## 6. Concluding Remarks

The color of plant food is an economic trait and is determined by the contents and varieties of pigments. Anthocyanins, as an important natural pigment, can give food attractive colors. In consideration of the health-promoting value of anthocyanins, foods rich in anthocyanins are more likely to capture consumers’ interest. It is of great importance to produce foods have an abundance of anthocyanins. Various transgene stacking and CRISPR/Cas systems can be selected to achieve plant genome modification and produce anthocyanin-enriched crops. However, transgenic or gene-edited foods are subject to strict legal regulations before marketing, and the relevant laws in most countries and regions may be different or imperfect, thus limiting the development and application of those traditional genome-modified techniques. To address this issue, transgene or marker-free technologies are developed and applied in crop genome improvement in recent years. However, more modifications may be needed when we use these tools across species. It is hoped that more genome-modified crops with improved anthocyanins but without foreign genes can be produced and accepted by the market and consumers.

## Figures and Tables

**Figure 1 molecules-28-00866-f001:**
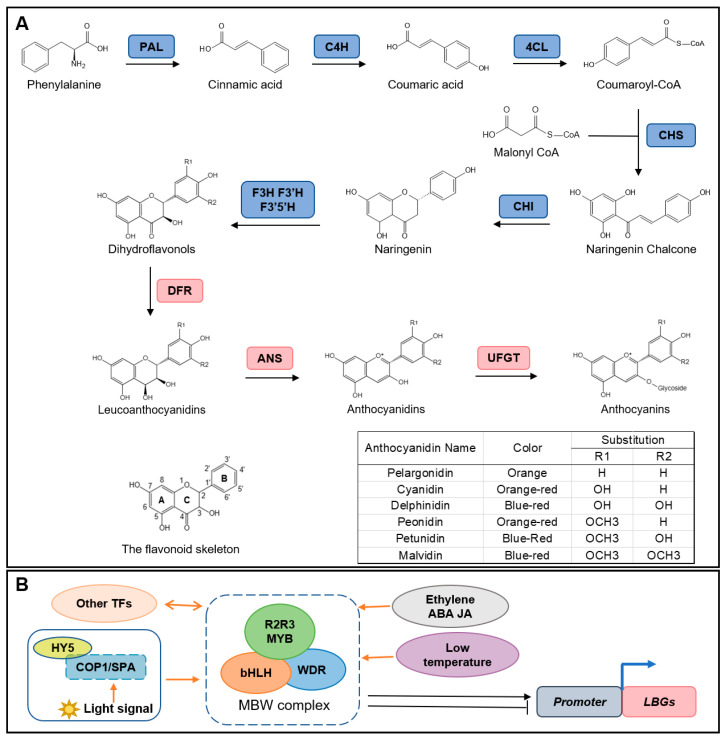
Anthocyanin biosynthesis pathway and the regulation factors. (**A**) Schematic pathway of anthocyanin biosynthesis in plants and the chemical structure of anthocyanins. Blue and light-red boxes represent *EBGs* and *LBGs*. (**B**) Other factors regulate anthocyanin accumulation in plants through affecting the MBW complex. The key *LBGs DFR*, *ANS* and *UFGT* are mainly regulated by a conserved MYB-bHLH-WDR (MBW) complex. The MBW complex interacts with other transcriptional factors and is regulated by phytohormones including ethylene, ABA (abscisic acid) and JA (jasmonic acid). The environment factors temperature and light signal affect the activity of the MBW complex. PAL, phenylalanine ammonia lyase; C4H, cinnamate 4-hydroxylase; 4CL, cinnamic acid-4-hydroxylase; CHS, Chalcone synthase; CHI, chalcone isomerase; F3H, flavanone-3-hydroxylase; F3′H, flavanone 3′-hydroxylase; F3′5′H, flavanone 3′5′-hydroxylase; DFR, dihydroflavonol 4-reductase; ANS, anthocyanidin synthetase; UFGT, flavonoid 3-O-glucosyltransferase.

**Figure 2 molecules-28-00866-f002:**
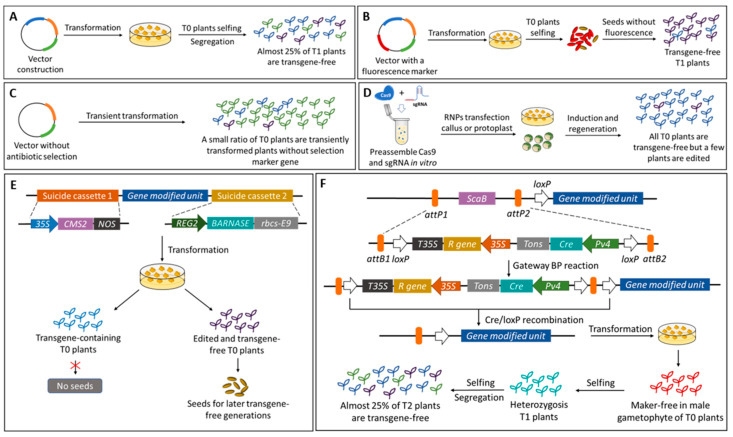
Strategies for producing transgene or marker-free crops that are rich in anthocyanins. (**A**) Eliminating foreign sequences through genetic segregation. T0 plants are usually heterozygous and with transgenes. (**A**) A small number of T1 generation segregating from T0 plants will be transgene-free and the modified locus will be homozygous. (**B**) Fluorescence tag-assisted screening. By fusion, a fluorescence marker such as *mCherry* or GFP, seeds without foreign constructs, can be easily identified. (**C**) Using transient expressing editor without antibiotic selection. Without biotic selection, *Agrobacterium*-mediated transient expression of foreign genes but there are no transgenes integrated into the plant genome. (**D**) Preassembled CRISPR/Cas9 ribonucleoproteins (RNPs). Cas9 protein and sgRNA are preassembled in vitro, then the RNP complex is used to transfect callus or protoplast. Thus, all the T0 generation is transgene-free. (**E**) Using suicide Transgene Killer CRISPR (TKC) system. There are two suicide units assembled on the plasmid: the expression of a rice male gametophyte specific lethal gene *CMS2* is driven by *CaMV 35 S*, and another lethal gene *BARNASE* is driven by a rice early embryo-specific promotor *REG2*. In T0 plants, transgene-containing pollen and embryos are killed. (**F**) Multi-transgene stacking systems with a marker-free donor vector. In the donor vector plasmid, Cre recombinase is driven by a pollen-specific promotor *Pv4*. The *Cre* and resistance gene (*R gene*) cassette is assembled into the transgene plasmid by Gateway BP reaction. Then Cre expression in pollen mediates the self-elimination of *Cre* and *R* gene cassette based on the Cre/loxP recombination. Marker-free progeny can be screened by genetic segregation in the T2 generation.

**Table 1 molecules-28-00866-t001:** Representative anthocyanins in common plant food.

Species	Part for Determination	Representative Compounds	Reference
*Fragaria × ananassa*	Receptacle	Pelargonidin-3-*O*-glucosides	[31]
*Rubus*	Fruit	Cyanidin-3-*O*-glucosides	[28]
*Vaccinium*	Fruit	Delphindin-3-*O*-galactoside, cyanidin-3-*O*-glucoside, petunidin-3-*O*-glucoside, malvidin-3-*O*-galactoside	[31]
*Ribes nigrum*	Fruit	Delphinidin-3-*O*-glucoside, delphinidin-3-*O*-rutinoside	[28]
*Vitis vinifera*	Peel	Delphinidin-3-*O*-glucoside, malvidin-3-*O*-glucoside, petunidin-3-O-glucoside, malvidin-3-*O*-glucoside-5-*O*-glucoside	[29]
*Citrus sinensis*	Pulp	Cyanidin-3-*O*-glucoside, cyanidin-3-*O*-(6’’-malonylglucoside)	[32]
*Lycium ruthenicum*	Fruit	Petunidin-3,5-*O*-diglucoside	[33]
*Oryza sativa*	Seed	Cyanidin-3-*O*-glucoside, peonidin-3-*O*-glucoside	[34]
*Zea mays*	Fruit	Cyanidin-3-*O*-glucoside, pelargonidin-3-*O*-glucoside, peonidin-3-*O*-glucoside, cyanidin-3-*O*-(6″-malonylglucoside), pelargonidin-3-*O*-(6″-malonylglucoside), peonidin-3-*O*-(6″-malonylglucoside)	[30]

## Data Availability

Not applicable.

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
