# Peer review of "Anthocyanins in Plant Food: Current Status, Genetic Modification, and Future Perspectives"

_molecules, 2023, doi:10.3390/molecules28020866_

Round 1

Reviewer 1 Report

The authors have done a detailed review of plant-based anthocyanins, their importance as food and the future use of genetic modification including genome editing to alter anthocyanin content. The only correction I would like to suggest is in table 1, where they should include an additional column to indicate what part of the plant species actually used as a source of anthocyanins.

Author Response

REVIEWER 1

The authors have done a detailed review of plant-based anthocyanins, their importance as food and the future use of genetic modification including genome editing to alter anthocyanin content. The only correction I would like to suggest is in table 1, where they should include an additional column to indicate what part of the plant species actually used as a source of anthocyanins.

Response 1: Many thanks for your suggestion, and we have added the information about the exact part for determination in Table 1.

Reviewer 2 Report

The Manuscript (MS) by Peiyu Zhang and Hongliang Zhu entitled“Anthocyanins in plant food: current status, genetic modification, and future perspectives” presented a very well-described review of summarized results of the studies of anthocyanins with high antioxidant activities in different plant species, their biosynthesis, advances in the molecular regulation and the progress in using the CRISPR/Cas gene editing or multigene overexpression methods to improve plant food anthocyanins content and promote human health.

The manuscript includes 110 references and described the majority of recent studies on the molecular regulation of the biosynthesis of plants anthocyanins, genetic engineering techniques, and future perspectives. The work is scientifically sound and comprehensively described. I consider that minor revision is required before the MS can be accepted for publication.

 1.  “Introduction” needs more information about the diversity of anthocyanins in plants related to different colors of plant food and the molecular regulation of their biosynthesis.

2.  In 2.2 the authors didn´t present full scientific and Latin names of the plant species and later shortly in the Tables (for example grape Vitis vinifera in the text, V. vinifera in Table 1 and Table 2). The same corrections needs for the whole text when the species are presented the first and the next time.

3.  In 2.2 it is suggested to present the concentration of the anthocyanins in the same volume.

4.  In Table 2 it is recommended to add information about transcription factors involved in anthocyanin biosynthesis in grapes (several MYB transcription factors as one of the most studied plants with anthocyanins diversity) and other plants from Table 1.

5.  In 4.2 for a broader view it is recommended to include references about other anthocyanin-improved transgenic crops for example from the publication https://www.mdpi.com/1422-0067/22/16/8752

6.  References should be described according to the rules of the Journal with

6a Authors must be listed in the same order as they appear in the original document. Different authors are separated with semicolons (‘;’) as in the example: Author 1; Author 2; Author 3;

6b For papers co-authored by a large number of persons (more than 10 authors), you can either cite all authors or cite the first ten authors, then add a semicolon and add ‘et al.’ at the end as in the example: Author 1; Author 2; Author 3; Author 4; Author 5; Author 6; Author 7; Author 8; Author 9; Author 10; et al.

Author Response

REVIEWER 2

  1. “Introduction” needs more information about the diversity of anthocyanins in plants related to different colors of plant food and the molecular regulation of their biosynthesis.

Response 1: Thanks for this suggestion, and we have added the information as you mentioned in the “Introduction” part.

  1. In 2.2the authors didn´t present full scientific and Latin names of the plant species and later shortly in the Tables (for example grape Vitis vinifera in the text, V. vinifera in Table 1 and Table 2). The same corrections needs for the whole text when the species are presented the first and the next time.

Response 2: Many thanks for your suggestion. We have checked and revised the Latin names of the plant species in the text and table.

  1. In 2.2it is suggested to present the concentration of the anthocyanins in the same volume.

Response 3: Thanks for your suggestion, and we unified the unit as mg 100 g-1.

  1. In Table 2 it is recommended to add information about transcription factors involved in anthocyanin biosynthesis in grapes (several MYB transcription factors as one of the most studied plants with anthocyanins diversity) and other plants from Table 1.

Response 4: We have added information about MYBs involved in grape anthocyanin synthesis according to recently published references, and other plant species in Table 2.

  1. In 4.2 for a broader view it is recommended to include references about other anthocyanin-improved transgenic crops for example from the publication https://www.mdpi.com/1422-0067/22/16/8752

Response 5: Thank you for reviewing our manuscript, and we have tried to add more information about anthocyanin-improved transgenic crops in the 4.2 section. But there are limited positive examples of anthocyanin-improved transgenic crops in existing studies, and we do not cite those examples of negative effects.

  1. References should be described according to the rules of the Journal with

6a Authors must be listed in the same order as they appear in the original document. Different authors are separated with semicolons (‘;’) as in the example: Author 1; Author 2; Author 3;

6b For papers co-authored by a large number of persons (more than 10 authors), you can either cite all authors or cite the first ten authors, then add a semicolon and add ‘et al.’ at the end as in the example: Author 1; Author 2; Author 3; Author 4; Author 5; Author 6; Author 7; Author 8; Author 9; Author 10; et al.

Response 6: Many thanks for your reminder. We have corrected the format of the references in the revised manuscript.

Reviewer 3 Report

1) English should be checked and proofread for the correct use of articles, adjectives versus adverbs, adjectives versus nouns, plural versus singular, present versus past tense etc.

2) The term "aglycone" (which is actually the flavonoid with the glycan attached) is misused, presumably due to a misunderstanding, to designate the glycan moieties (groups) covalently bound to the flavone core molecule.

3) There are miscomprehensions of risk and regulation at various places in the manuscript:

-3a) The text at various places alludes to the use of transgene-free techniques as a way to allay concerns over the toxicity and allergenicity of the GM crops (wrongfully implying that transgenic crops are risky).  Tis in my view obfuscates the facts that 1) transgenic produces undergo a pre-market regulatory safety assessment in many countries in this world according to an internationally harmonized safety assessment approach as recommended by FAO/WHO Codex Alimentarius guidelines; 2) the reason why so many Chinese researchers apparently would prefer to use DNA-free methods is because these are exempt from the regulations on genetically modified organisms under their national law (which is different from that elsewhere)

-3b) It states that PPO-silenced mushrooms were commercialized in the USA and fell outside regulation:  This is inaccurate: these have never been commercialized and are just one out of many other genetically engineered crops which had been notified under the "Am I regulated" procedure (in the meantime replaced by the SECURE ruling) to the USDA.  It would be better to omit this example.

4) Structure:

-4a) There is an imbalance in the paper's distribution of attention: there is a lot detail regarding the various mechanisms of regulation of flavonoid metabolism and the possibility (very generic) to use non-transgenic methods to edit crops, whilst the example of GM crops with altered flavonoid metabolism is marginal. Aren't their more examples of gene-edited crops with altered flavonoid metabolism?  Why isn't the conventionally bred purple tomato Indigo Rose (previously cited) not compared with the transgenic one described in 4.2.1?  Please also refer to the regulatory status review by the USDA of the transgenic purple tomato with Del1/Rosea modifications (see RSR Tomato.pdf (usda.gov), June 2022)

-4b) In addition, the narrative explanation would merit from a bit more of graphic explanation:  For instance, what is the structure of the various intermediary compounds formed by the various enzymes mentioned?

-4c) Why aren't the only commercial GM plants with altered flavonoid metabolism, namely flowers (carnation, rose, chrysanthemum) from the Suntory/Florigene company mentioned at all?

5) When describing genes, also provide the host species (e.g., potato Solanum tuberosum for the genes that have their names starting with "St...", tomato for Sl...)

6) The anti-cancer properties referred to in line 289 is a bold claim that should be supported by hard evidence or otherwise removed from the text 

Author Response

REVIEWER 3

1) English should be checked and proofread for the correct use of articles, adjectives versus adverbs, adjectives versus nouns, plural versus singular, present versus past tense etc.

Response 1: Many thanks for your review of our manuscript, and we have checked and corrected the words and sentences of the full text.

2) The term "aglycone" (which is actually the flavonoid with the glycan attached) is misused, presumably due to a misunderstanding, to designate the glycan moieties (groups) covalently bound to the flavone core molecule.

Response 2: We are sorry for the misuse here, and we have corrected this in the revised manuscript (Section 2.1 and Figure 1).

3) There are miscomprehensions of risk and regulation at various places in the manuscript:

-3a) The text at various places alludes to the use of transgene-free techniques as a way to allay concerns over the toxicity and allergenicity of the GM crops (wrongfully implying that transgenic crops are risky).  Tis in my view obfuscates the facts that 1) transgenic produces undergo a pre-market regulatory safety assessment in many countries in this world according to an internationally harmonized safety assessment approach as recommended by FAO/WHO Codex Alimentarius guidelines; 2) the reason why so many Chinese researchers apparently would prefer to use DNA-free methods is because these are exempt from the regulations on genetically modified organisms under their national law (which is different from that elsewhere)

Response 3a: Many thanks for your review. We deleted some inappropriate sentences and highlight the important role of legal regulation in the production and marketing of GM crops in the revised manuscript (such as line 61-71). Although many international organizations and countries have adopted the form of legislation to implement biosecurity management, sometimes the modern biotechnology breeding is limited by the imperfect regulatory system (such as in China). In thus cases, we believe that transgene/marker-free technology is promising and worth further exploring.

-3b) It states that PPO-silenced mushrooms were commercialized in the USA and fell outside regulation:  This is inaccurate: these have never been commercialized and are just one out of many other genetically engineered crops which had been notified under the "Am I regulated" procedure (in the meantime replaced by the SECURE ruling) to the USDA.  It would be better to omit this example.

Response 3b: Thanks for this suggestion, we have omitted this example in the revised manuscript.

4) Structure:

-4a) There is an imbalance in the paper's distribution of attention: there is a lot detail regarding the various mechanisms of regulation of flavonoid metabolism and the possibility (very generic) to use non-transgenic methods to edit crops, whilst the example of GM crops with altered flavonoid metabolism is marginal. Aren't their more examples of gene-edited crops with altered flavonoid metabolism?  Why isn't the conventionally bred purple tomato Indigo Rose (previously cited) not compared with the transgenic one described in 4.2.1?  Please also refer to the regulatory status review by the USDA of the transgenic purple tomato with Del1/Rosea modifications (see RSR Tomato.pdf (usda.gov), June 2022)

Response 4a: Many thanks for reviewing our manuscript. Although there are many examples of using CRISPR/Cas system to knock out related genes in various plant food species to regulate anthocyanin biosynthesis, but most of the genes are positive factors and the gene editing vents usually showed negative effects on the accumulation of anthocyanin in food (Kumar et al., 2022). In existing studies, there are limited examples of classical gene editing methods to improve anthocyanin content in plant-based foods, so we do not cite these examples of negative effects. And we have discussed the Indigo Rose cultivar compared with the transgenic one in section 4.2.1 (line 324-326). And we have added the information about the regulatory status review by the USDA of the transgenic purple tomato with Del1/Rosea modifications (line 303-307).

[Kumar D, Yadav A, Ahmad R, Dwivedi UN, Yadav K. CRISPR-Based Genome Editing for Nutrient Enrichment in Crops: A Promising Approach Toward Global Food Security. Front Genet. 2022 Jul 14;13:932859. doi: 10.3389/fgene.2022.932859.]

-4b) In addition, the narrative explanation would merit from a bit more of graphic explanation:  For instance, what is the structure of the various intermediary compounds formed by the various enzymes mentioned?

Response 4b: Thanks for your suggestion, and we have improved the figure 1 in the revised manuscript.

-4c) Why aren't the only commercial GM plants with altered flavonoid metabolism, namely flowers (carnation, rose, chrysanthemum) from the Suntory/Florigene company mentioned at all?

Response 4c: Thank you for reviewing our manuscript. Because this review is focused on the plant eaten food, so we didn’t discuss other kinds of plant species such as the flowers you mentioned.

5) When describing genes, also provide the host species (e.g., potato Solanum tuberosum for the genes that have their names starting with "St...", tomato for Sl...)

Response 5: Many thanks for your kind reminder, and we have corrected this in the revised manuscript (such as Table 1, line 325-326).

6) The anti-cancer properties referred to in line 289 is a bold claim that should be supported by hard evidence or otherwise removed from the text 

Response 6: Many thanks for this suggestion, and we removed this sentence from the text.

Round 2

Reviewer 3 Report

Thanks to the authors for having taken on board my comments.  There is one major and are a few minor editorial details, as follows:

Major comment: the discussion (lines 486-487) still imply that transgene-free methods are to address safety issues over gene-edited foods (whereas this has already been removed from other parts in response to my previous comment).  This can be deleted as well

Detailed comments:

Figure 1: "Coumaric acid" instead of "Coumalic acid"

Line 288: insert "are" after "sites"

Line 305: delete "in 2008": this is redundant and confusing

Line 482:  insert "of" after "consideration"

Line 492: replace "but do not contain" with "without"

Author Response

Response to Reviewer 3 Comments

REVIEWER 3

Major comment: the discussion (lines 486-487) still imply that transgene-free methods are to address safety issues over gene-edited foods (whereas this has already been removed from other parts in response to my previous comment).  This can be deleted as well

Response 1: Many thanks for your suggestion. We changed this sentence as follows: “However, transgenic or gene-edited foods are subject to strict legal regulations before marketing, and the relevant laws in most countries and regions may be different or imperfect, thus limiting the development and application of those traditional genome-modified techniques.” (line 486-489)

Detailed comments:

Figure 1: "Coumaric acid" instead of "Coumalic acid"

Response 2: We have corrected this in figure 1.

Line 288: insert "are" after "sites"

Response 3: We have inserted “are” after “sites” here.

Line 305: delete "in 2008": this is redundant and confusing

Response 4: We have deleted “in 2008” in line 305.

Line 482:  insert "of" after "consideration"

Response 5: We have inserted “of” after “consideration” (line 482).

Line 492: replace "but do not contain" with "without"

Response 6: We have replaced “but do not contain” with “without” (line 494).